# Evaluation and Characterization of Ultrasonic Cutting of Monofilament Nylon

Hainan Zhang, Zhihuang Shen *, Zhonghe Cao, Dapan Hou, Tao Jiang and Xianzhen Ye

School of Marine Equipment and Mechanical Engineering, Jimei University, Xiamen 361021, China; jmuzhn@163.com (H.Z.); 18855039055@163.com (Z.C.); dphou@jmu.edu.cn (D.H.); todd_jmu@126.com (T.J.); yexz2015@jmu.edu.cn (X.Y.)
* Correspondence: jmuszh@163.com

**Abstract:** Nylon (polyamide) is a typical composite base material, but its low hardness, poor rigidity, and low dimensional stability easily lead to burrs, fractures, curling, and other problems after nylon end faces are processed by conventional techniques. To prevent these problems, this article conducted experimental research comparing ultrasonic and conventional cutting of monofilament nylon and used the controlled variable method to analyze the micromorphology after cutting at different speeds and tensions. A novel ratio was also derived to characterize the topographic quality, and the feasibility and advantages of ultrasonic cutting of nylon were verified by evaluating the nylon surface quality under different cutting conditions. Test results showed layering on the rear face of nylon with ultrasonic cutting as well as a morphology quality directly proportional to the feed speed of the tool and inversely proportional to the tensile force exerted on the material. The end face of nylon after ultrasonic cutting provided a ratio 2.9% lower than that obtained from conventional cutting as well as a roughness reduction of 31.3%. Compared with conventional cutting, the end face was preserved, and no large-area breakage or edge curling occurred when using ultrasonic cutting. Moreover, a force analysis theoretically confirmed the advantages of ultrasonic cutting.

**Keywords:** ultrasonic special cutting; nylon material; micromorphology





## 1. Introduction

Nylon (polyamide) is the base material of many modern composite materials. It is widely used in the automobile, aerospace, electronics, and electrical industries owing to its high toughness, good elasticity, high softening point, low friction coefficient, wear resistance, and high mechanical strength. Its applications include chemicals, bulletproof materials, and biomedical solutions [1–3]. Yu Yuan et al. [4] analyzed the relationship between the incision morphology of nylon filament, its curled length after cutting, and the sharpness of the shaver. The results show that the notch morphology of nylon filament can be divided into three regions, and the characteristic morphology is closely related to the crimp length of nylon filament. Jagtap T et al. [5] studied that feed rate is the most important factor affecting the surface roughness and material removal rate when turning on nylon 6 material, and the optimal processing parameter combination is proposed. Nguyen HC et al. [6] investigated the cutting characteristics and thermal dependency of nylon-polyethylene laminated film under wedge indentation, revealing that blade tip thickness significantly affects the film's bending angle and cutting quality while also highlighting the impact of blade temperature on the film's deformation. Park YH et al. [7] introduced a nylon wire cutting-type holding and release mechanism for a 6U CubeSat's solar panel, enhancing holding capability and simplifying the tightening process of the nylon wire. Suksawat B [8] studied the effect of cutting conditions on the main cutting force and surface roughness in the turning of cast nylon and found that an increase in cutting speed and depth of cut increases the main cutting force, while the surface roughness is affected by an increase in feed rate and a decrease in cutting speed.

For processing composite materials, Tangwarodomnukun et al. [9] used water-cooled laser processing of carbon fiber-reinforced polymers, thereby reducing the amount of applied laser energy and thermal damage while improving processing. For composite materials in different applications and with various compositions, special processing methods are required to obtain topographic characteristics such as grooves, chamfers, and end faces. Ozkan et al. [10] used TiAlN- and TiN-coated tools to mill carbon fiber-reinforced polymers and studied the influence of cutting on the surface roughness and applied force. Similarly, ultrasonic machining can be used to accurately cut composite materials such as ceramics, semiconductors, glass fibers, carbon fibers, and honeycomb materials [11]. Huang et al. [12] applied fracture mechanics to introduce the dynamic stress intensity in the study of microscopic fracturing of honeycomb composites subjected to ultrasonic cutting. In addition, they theoretically analyzed the ultrasonic cutting of honeycomb composites. Zhang et al. [13] and Sun et al. [14] used ultrasonic cutting for Nomex and aluminum honeycomb cores, finding that ultrasonic cutting can substantially reduce the cutting force and improve the cut surface. Shu et al. [15] used rotary ultrasonic milling of carbon fiber composites and found that ultrasonic vibration can reduce pores and cracks on the machined surface.

The abovementioned studies investigated the influence of tool amplitude and cutting speed on the morphology after cutting composite materials under ultrasonic vibration. In addition, the performance of composite materials was qualitatively evaluated by observing the processing topography regarding aspects such as edge burrs and surface quality. Cutting monofilament nylon is a key technology that is critical to improving the performance and range of applications of composites. By precisely controlling the size and shape of monofilament nylon, the mechanical strength, wear resistance, and corrosion resistance of composites can be significantly improved. Fine monofilament nylon cutting technology is the key to achieving lightweight and high-performance design. However, scarce research is available on cutting characteristics. In this paper, ultrasonic cutting is introduced and the cutting quality of nylon monofilament is evaluated, which is the greatest innovation different from previous studies. Specifically, monofilament nylon was cut using a knife (conventional cutting) and an ultrasonic knife (ultrasonic cutting), and the micromorphology of the end face was observed and studied. The end face microtopography of the two cutting methods was compared, in addition to the influence of cutting parameters on the surface quality. In addition, a method for evaluating the cutting quality of processed monofilament nylon was derived. Finally, the feasibility of ultrasonically cutting monofilament nylon was verified by considering the cutting force theory.

## 2. Monofilament Nylon Cutting

For ultrasonic cutting, the relative position and feeding direction of the cutter and nylon were set as shown in Figure 1. The cutting nylon test used a straight-edge blade, which is the ultrasonic attached to the tool. The blade and nylon were axially perpendicular; that is, the cutting direction was perpendicular to the table. For monofilament nylon cutting, two different cutting methods were evaluated. The first condition considered sample straightening. Specifically, the table was considered to determine distance A and measure a mark with the same length for the nylon sample. Then, the end of the nylon sample was tied to a fixed post. This article used the winding bundle method to achieve force transmission along the nylon axis. The other end was attached to a weight such that the nylon sample exactly matched distance A and was straightened. The second force condition considered a range of monofilament nylon tensile strength to apply a force greater than that exerted by straightening using the weight. The range was obtained by changing the mass of the weight to apply tension to the nylon sample.

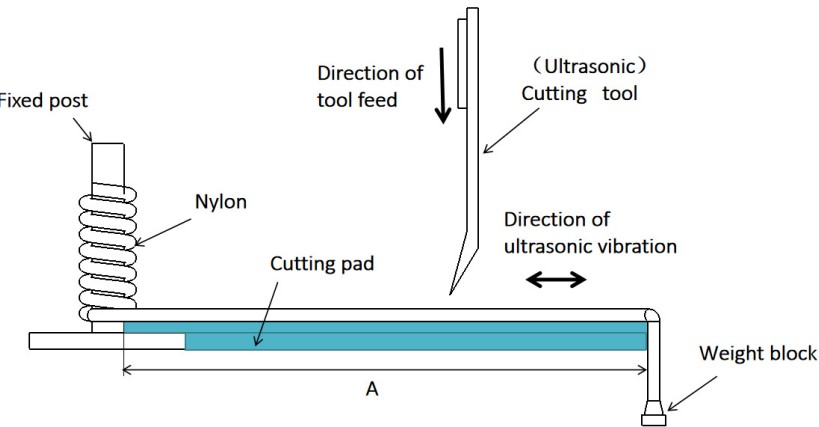

**Figure 1.** Schematic diagram of ultrasonic cutting.

### 3. Cutting Force Analysis

Ultrasound is superimposed along the $X$, $Y$, and $Z$ axes of the tool, and the resulting high-frequency vibration accelerates material processing to achieve efficient cutting, as shown in Figure 2. When the tool cuts along the $Z$-axis, cutting force $F_R$ mainly comprises the normal force $F_N$ of the material on the blade face, friction force $F_f$ in the parallel direction of the blade face, and resistance $F_r$ to the blade motion. In addition, $F_N$ is the resultant force of material extrusion force $F_e$ and $F_r$, and the friction coefficient of quenched steel and polyamide nylon is above 0.3 without lubrication. As the contact surface between the blade and nylon is small, nylon is considered a low-stiffness material, and the cutting force is mainly generated by the frictional resistance between the material and tool, leaving only force $F_N$.

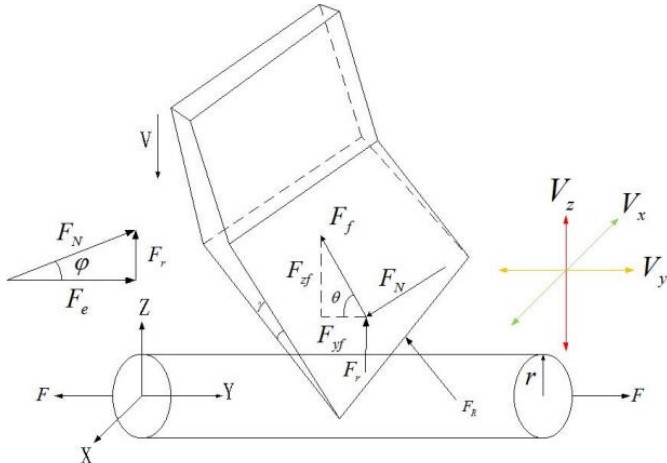

**Figure 2.** Schematic diagram of ultrasonic cutting method.

When ultrasound is applied along the feeding direction ($Z$-axis) of the tool, the resistance of the tool reduces along that direction without notably altering the cutting efficiency. When ultrasound is applied along the $X$-axis, the contact between the tool and nylon increases during cutting. The area increases the frictional force, which undermines cutting. When ultrasound is applied along the $Y$-axis, the cutting rake angle $\alpha$ is in a dynamic state of $\alpha_1 - \alpha_2 - \alpha_1$ during cutting ($\alpha_1 < \alpha_2$). This state is more conducive to the destruction and removal of materials. Therefore, this study uses ultrasonic tool vibration along the nylon axial.

To determine the factors influencing the cutting quality, the tool force should be analyzed. To simplify the analysis, only the force generated by the feed speed of the tool is considered for conventional cutting. The simplified schematic diagram of the force for

conventional cutting is shown in Figure 3. Ultrasonic cutting increases the vibration of the tool along the *X*-axis, depending on the force of conventional cutting. The vibration displacement equation can be expressed as follows:

$$x = B \sin(2\pi\, ft)$$

where B represents the amplitude and *f* represents the frequency of ultrasonic vibration. The tool feeds along the *Y*-axis. When the blade cuts inside the nylon sample, the sides of the tool are subjected to friction forces $f_1, f_2$ from the nylon parallel to the blade face and to pressures $F_{N1}$, $F_{N2}$ perpendicular to the blade face and feed [15]. Pressure $F_r$ acts on the blade in the opposite direction (Figure 4a). When the cutter moves tangentially to the nylon sample at frequency *f*, the sample is first squeezed by the blade. As nylon is a thermoplastic material, the material follows the extrusion process. The blade curls along the feeding direction and the extrusion force of the tool surpasses the strength limit of nylon over time, destroying the material (Figure 4b). Cutting after the damage is mainly destroyed by the tool force. In a later cutting stage, the tool and pulling forces act simultaneously and tearing caused by the pulling force dominates over time. Finally, nylon is pulled and torn by the action of the pulling force, leaving the edge collapsed and buried (Figure 4c).

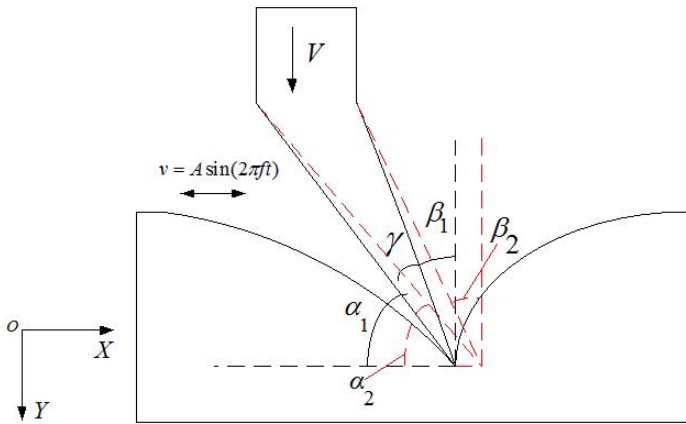

**Figure 3.** Dynamic schematic diagram of ultrasonic cutting.

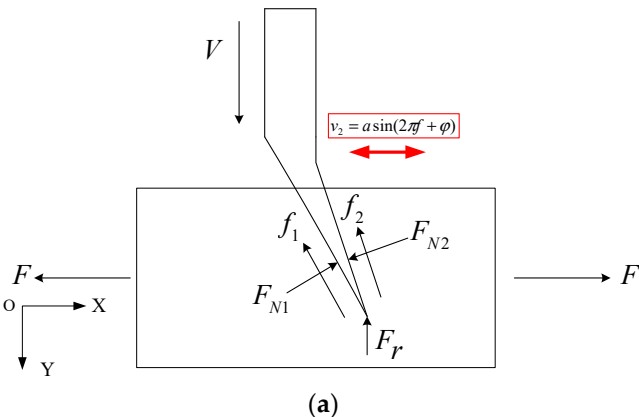

(**a**)

**Figure 4.** *Cont*.

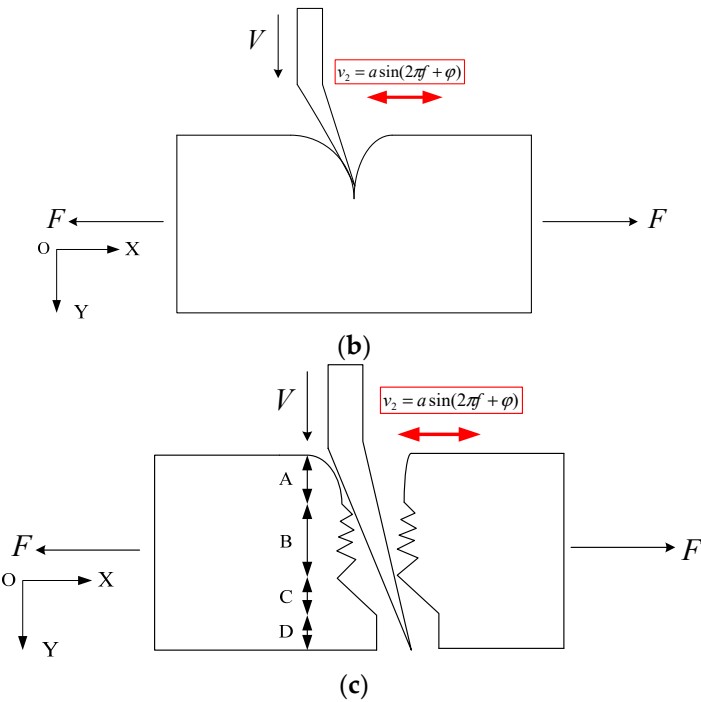

**Figure 4.** Simplified diagram of cutting process force schematic (**a**–**c**).

During ultrasonic cutting, the ultrasonic vibration is sinusoidal with respect to the *X*-axis, accelerating the initial damage of the blade to the nylon surface and slowing down the inward curling of nylon owing to extrusion, thereby obtaining a higher-quality cut surface. Increasing the feed speed of the tool under a given tension can increase the contact time between the tool and the nylon end face, thereby reducing the end face roughness. Hence, within the same cutting distance, the ultrasonic tool delivers an alternating force along the *X*-axis, which acts on the workpiece and delivers strong extrusion. The destruction of nylon is accelerated to complete cutting. Owing to the existence of the alternating force, a certain gap exists between the blade face and nylon, thus reducing the friction force *F*. For a constant feed speed of the tool, tension force *F* is applied to both ends of the nylon, and the component of *F* perpendicular to the blade surface reduces the force $F_N$ of the nylon on the tool, allowing the application of appropriate tension to mitigate curling.

## 4. Micromorphological Characteristics of Cut Nylon End Face

### 4.1. Monofilament Nylon Sample

This study used a monofilament cylindrical nylon sample, as shown in Figure 5, with the parameters listed in Table 1.

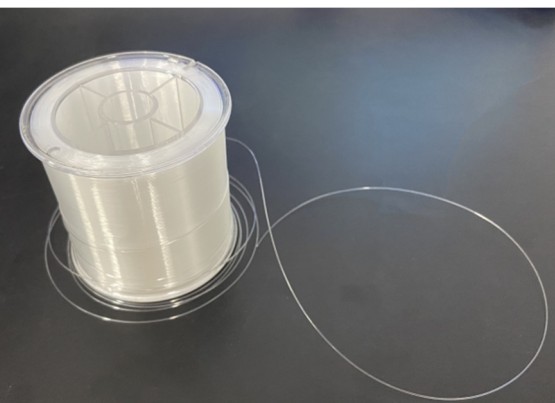

**Figure 5.** Physical diagram of nylon material.

**Table 1.** Nylon material parameters.

| Material | Diameter (mm) | Tensile Strength (MPa) | Density (g/cm³) | Elastic Modulus |
|----------|---------------|------------------------|------------------|-----------------|
| Nylon | 0.46 | 943.5 | 1.13 | 0.28 |

### 4.2. Cutting Test

The same tool (i.e., steel polished with 1000 CW sandpaper) was used for ultrasonic and conventional cutting. The tool was specially designed for ultrasonic vibration as input excitation. The cutting tool was mainly composed of piezoelectric ceramic sheets, electrode sheets, aluminum blocks, and steel sheets. The blade and nylon diameter directions were kept at 15° during cutting, as shown in Figure 6. In the test, the nylon sample was fixed to a post and stretched by a heavy block. The tension of nylon was lower than that of the true heavy block. To perform cutting at different speeds, the tool was connected to a power amplifier to achieve ultrasonic vibration cutting with the tool. The test bench is shown in Figure 7.

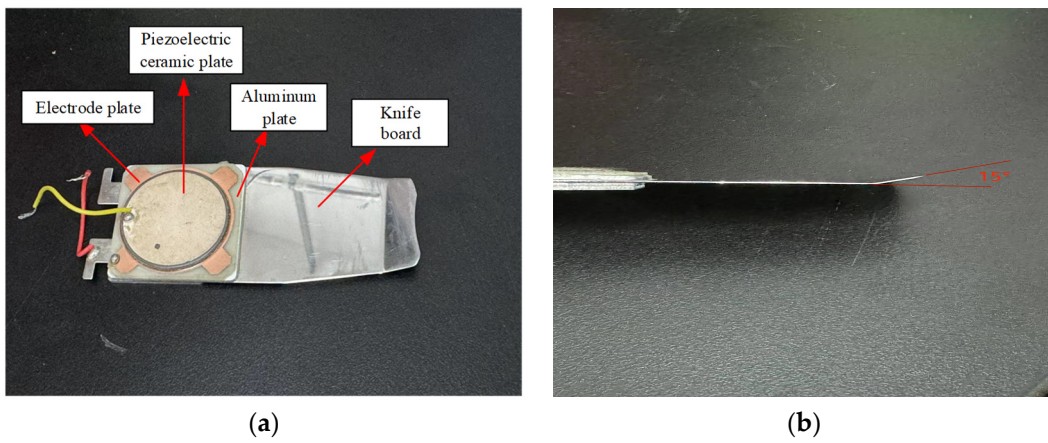

(**a**)      (**b**)

**Figure 6.** Physical drawing of the tool. (**a**) Main view of the tool; (**b**) Side view of the tool.

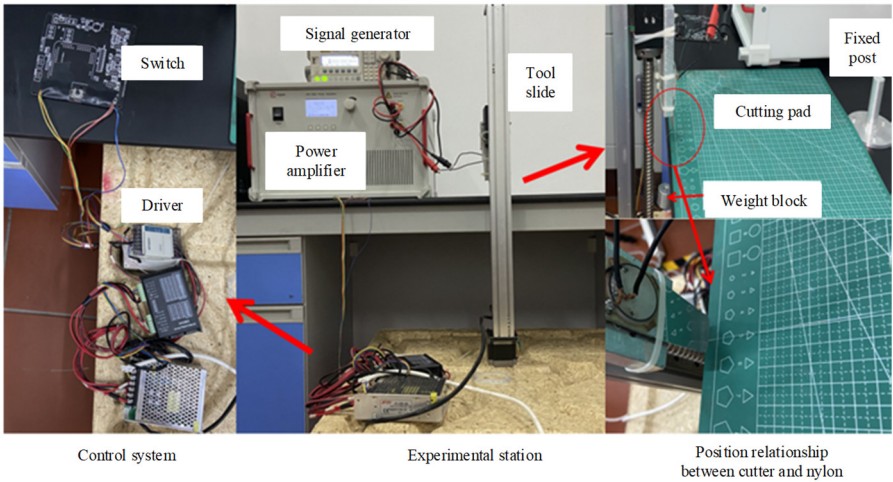

**Figure 7.** Cutting test chart.

The ultrasonic tool emitted the excitation signal through a signal generator and was increased by the power amplifier, and the tool vibration frequency was 24.75 kHz with an amplitude of 0.35 µm. The test used a DG1022V signal generator and ATA-3082 power amplifier manufactured by Xi'an Antai Electronic Technology, as well as an NLV-2500 laser vibrometer to detect the ultrasonic tool amplitude and KEYENCE VK-X1000 laser confocal microscope to observe the microscopic morphology of the cut nylon end face.

The morphological characteristics of nylon samples after cutting can be used as criteria for evaluating the cutting quality and feasibility of ultrasonic cutting. The observation of the end surface morphology of the cut nylon was analyzed regarding the end surface morphology properties and nylon edge morphology quality, where the nylon edge morphology was divided into tool-into-edge morphology and tool-off edge morphology.

### 4.3. Nylon Sample Morphology without Added Tension

To clearly observe the morphology of the nylon after conventional and ultrasonic cutting, tension was first applied to the monofilament nylon to reach a taut state, and the tool feed speed was 12.5 mm/s.

Figure 8 shows the nylon with a rolled edge toward the end face center at the knife edge, hardly maintaining the original shape of the nylon end face after cutting. The uneven force of the knife edge on the nylon surface under conventional cutting caused bright scratches on the end face owing to the convexity of the knife edge by hand polishing and residual debris on the edge of the end face. After ultrasonic cutting, the shape of the end face was substantially improved, as the rolled edge was mitigated, no significant bright scratches appeared, the overall shape remained intact, and the roughness was lower than that obtained after ordinary cutting.

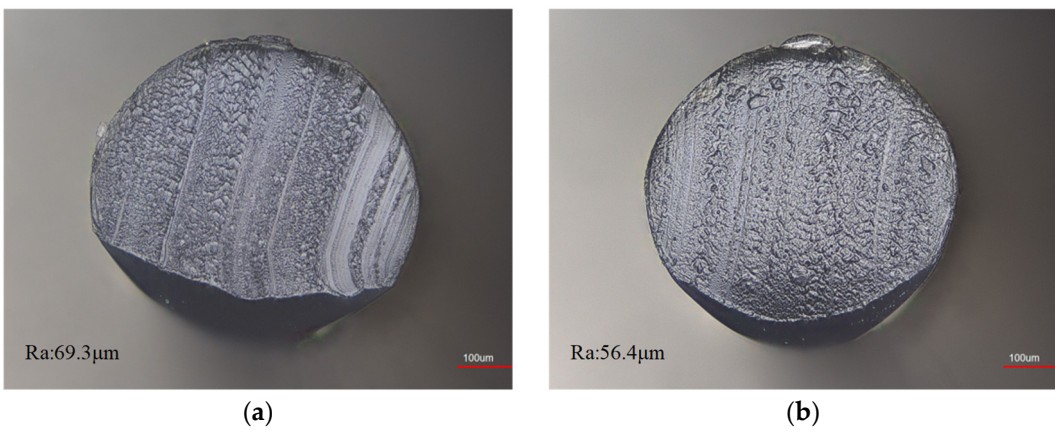

Ra:69.3µm 100um (a)

Ra:56.4µm 100um (b)

**Figure 8.** Microscopic view of cutting end face. (**a**) General cutting; (**b**) Ultrasonic cutting.

### 4.4. Micromorphology of Nylon with Added Tension

This study used a monofilament nylon sample. This material is often used in the fishing industry as a fishing line, and its surface softness and smoothness, as well as its ability to resist frictional damage from obstacles, determine the stability and usability of the product. To obtain an adequate cutting effect, this study evaluated the parameters listed in Table 2 for comparative cutting tests.

**Table 2.** Cutting nylon material test parameters.

| Cutting Method | Group | Tension (N) | Cutting Speed (mm/s) |
|---|---|---|---|
| General cutting | 1 | 17.15 | 3.125 |
| | | | 6.25 |
| | | | 12.5 |
| | | | 25 |
| | 2 | 12.25 | 12.5 |
| | | 14.7 | |
| | | 17.15 | |

**Table 2.** *Cont*.

| Cutting Method | Group | Tension (N) | Cutting Speed (mm/s) |
|---|---|---|---|
| Ultrasonic cutting | 1 | 17.15 | 3.125 |
| | | | 6.25 |
| | | | 12.5 |
| | | | 25 |
| | 2 | 12.25 | 12.5 |
| | | 14.7 | |
| | | 17.15 | |

(1)    Nylon end face morphological characteristics

When the nylon was subjected to a given tensile strength, cutting produced layering of the end face under conventional and ultrasonic cutting, as shown in Figure 9. Tool cutting first produced impact damage to the contact surface (segment A of damage). The tool feed movement produced blade force exerted on the material and material damage (segment B of damage). Nylon damage increased with lower tensile strength. Material damage by combined tensile and blade forces produced combined force damage (segment C of damage). As the nylon along the diameter direction of the damage deepened, cutting was completed by the tensile tear damage (segment D of damage). Overall, cutting was dominated by damage in segment B. Roughness $R_a$ of the four segments measured by confocal microscopy was 25.2 μm, 13.7 μm, 26.8 μm, and 32.8 μm for segments A–D, respectively. Thus, the increased motion for damage in segment B improved the cutting quality. Damage segment CD caused tensile tearing and a relatively large roughness.

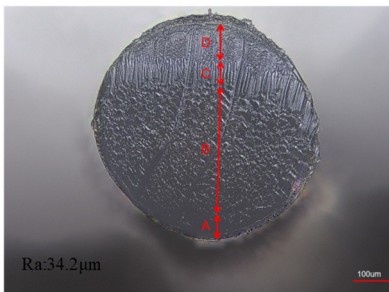

**Figure 9.** "Layering" diagram.

The ratio λ can be calculated to measure the quality of the end face cutting as follows:

$$\lambda = \frac{C + D}{A + B}$$

where *A*, *B*, *C*, and *D* indicate the distance of the line segment in Figure 8. As shown in Equation (2), a lower ratio indicates a higher cutting quality.

As nylon was straightened in the first test, cut layering was not obvious, and the rolled edge at the entrance of the tool was poor. Thus, this study also evaluated an initial feed rate of 12.25 mm/s to explore the effect of tension on the ratio indicating the cutting quality. For a given tool feed speed, the ratio tends to increase with increasing tension (Figure 10). When the nylon was straightened, the overall height difference of the end face under ultrasonic cutting was reduced by 2% compared with conventional cutting (Figure 11). At low tension (12.25 N) and feed speed (12.25 mm/s), curled edges appeared after conventional cutting, while they were substantially mitigated under ultrasonic cutting, which provided a reduced ratio. Overall, the ratio under ultrasonic cutting was reduced by 2.9%, and the roughness was reduced by 31.3% compared with conventional cutting.

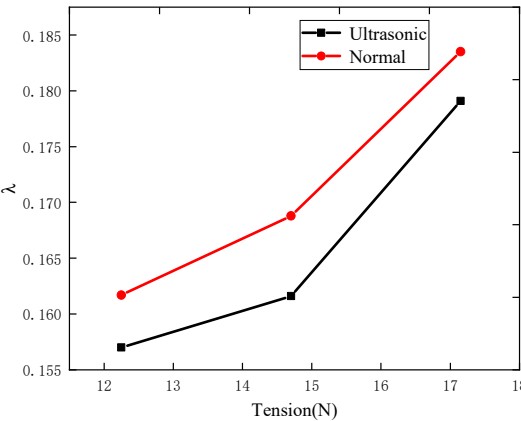

**Figure 10.** Comparison of ratio variation with tension for the same tool feed speed.

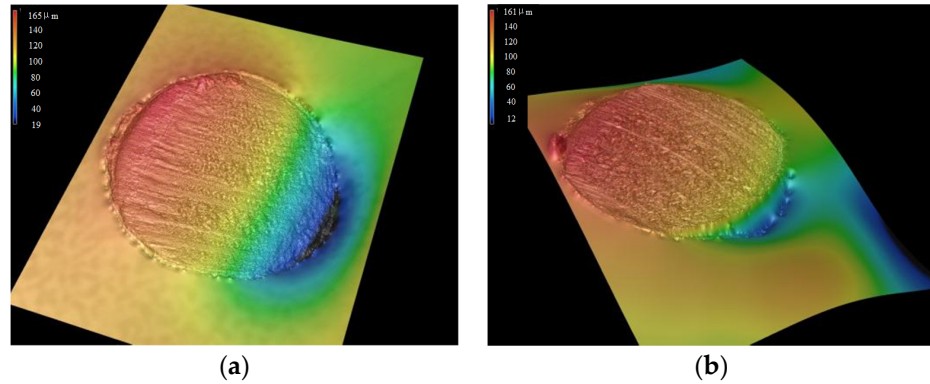

(**a**)                                                    (**b**)

**Figure 11.** Three-dimensional microscopic view of cutting under 12.25 N tension at 12.25 mm/s. (**a**) General cut 3D micrograph; (**b**) Ultrasonic cutting 3D microscopic view.

(2)    Nylon edge morphological characteristics

The feed direction of the tool and nylon diameter direction formed an angle of 15°, which could easily cause an uneven cutting end surface. This was mainly reflected in the damage to segment A, where the knife edge and nylon just contacted. The lower-end surface of conventional cutting proceeded under low feed speed and low tension. The height difference was 149 μm during conventional cutting and 146 μm during ultrasonic cutting, indicating mitigated curling under ultrasonic vibration (Figure 12). Edge collapse and burrs were likely to occur at some distance from the knife under conventional cutting, and as the tensile force increased, edge tear increased and reduced the cutting quality (Figure 13). On the other hand, tearing was reduced under ultrasonic cutting, and no obvious edge collapse was observed.

When nylon was pulled, it appeared to tear at the later stage of cutting. To explore the effect of the tool feed speed on the kerf, Figure 14 shows the relation between the feed speed and ratio under 17.15 N of tensile force applied at one end of the nylon sample. For the given tensile force, the ratio decreased with increasing feed speed, and the trends of conventional and ultrasonic cutting were similar. When the feed speed increased from 3.125 to 25 mm/s, the ratio for conventional cutting decreased from 0.1955 to 0.1812, representing a decrease of 7.3%, while that of ultrasonic cutting decreased from 0.1791 to 0.1570, representing a decrease of 12.3%. Thus, conventional and ultrasonic cutting exhibited similar patterns, while the ratio λ of the end face after ultrasonic cutting under the same cutting conditions was smaller, indicating improved cutting quality compared with conventional cutting.

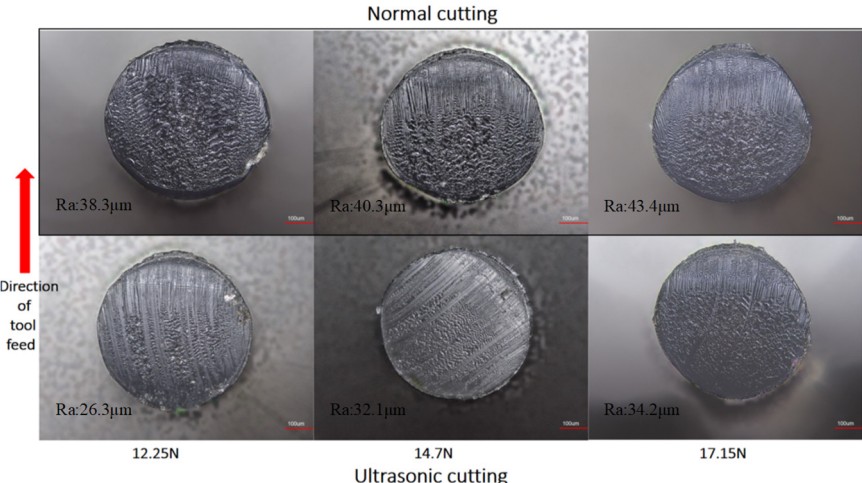

**Figure 12.** Comparison of nylon kerf shape with different tension at the same tool feed speed.

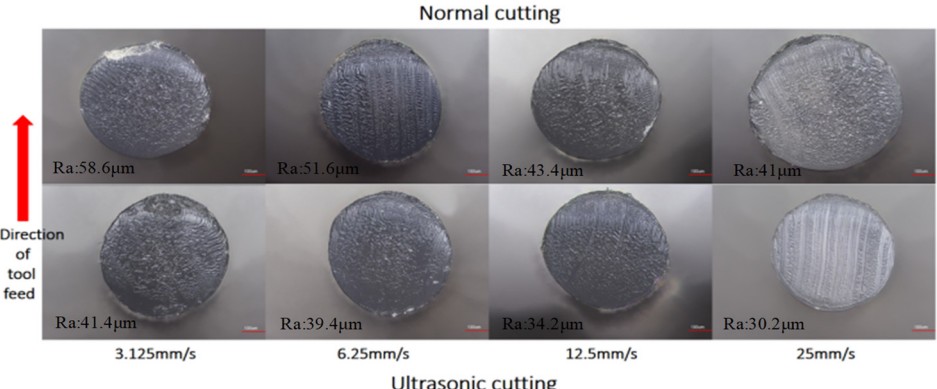

**Figure 13.** Comparison of microscopic morphology of nylon kerf with different feed speeds of tools under the same tension.

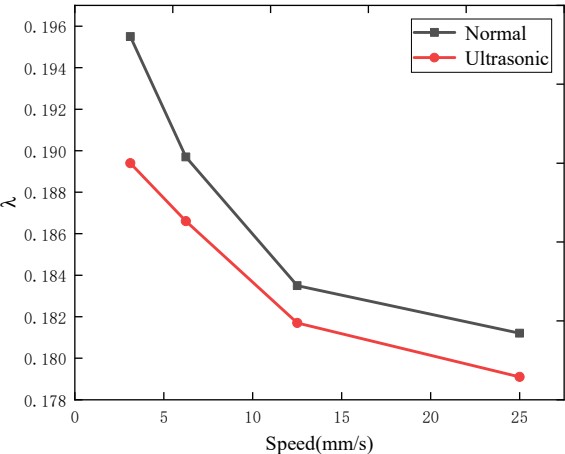

**Figure 14.** Comparison of ratio with speed under the same tension.

## 5. Conclusions

(1)   Conventional cutting is performed through a one-dimensional feed movement. Ultrasonic cutting involves small movements in a two-dimensional plane for a tool to complete the cutting of monofilament nylon samples and obtain high-quality cutting end faces, making it a feasible and convenient processing method.

(2) Ultrasonic cutting can overcome curling deformation caused by the extrusion of the tool. In addition, the edge shows no fracture or collapse, and the end face maintains a good shape without obvious deformation.

(3) Moreover, ultrasonic cutting can eliminate the dependence of the cutting quality on the position relationship between the cutter and nylon under appropriate tension. For a given tension, increasing the feed speed of the cutter can improve the cutting quality, with the increase in the ratio of the end face topography being 1.7 times compared with conventional cutting.

**Author Contributions:** Methodology, D.H.; Validation, Z.C.; Investigation, X.Y.; Data curation, T.J.; Writing—original draft, H.Z.; Writing—review & editing, Z.S. All authors have read and agreed to the published version of the manuscript.

**Funding:** This research was funded by Natural Science Foundation of Fujian Province (No. 2021J01853), Natural Science Foundation of Fujian Province (No. 2021J01850).

**Institutional Review Board Statement:** Not applicable.

**Informed Consent Statement:** Not applicable.

**Data Availability Statement:** Data are contained within the article.

**Conflicts of Interest:** The authors declare no conflict of interest.

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
