# Peer review of "Evaluation and Characterization of Ultrasonic Cutting of Monofilament Nylon"

_coatings, doi:10.3390/coatings14040462_

Round 1

Reviewer 1 Report

Comments and Suggestions for Authors

The manuscript under review explores innovative approaches to enhance the surface quality of nylon (polyamide), a widely used composite base material. The authors highlight the challenges associated with conventional cutting techniques, which often result in undesirable outcomes. To address these issues, an experimental research is presented comparing ultrasonic cutting to conventional methods.

A significant aspect of the study lies in the development of a novel ratio to characterize topographic quality, providing a quantitative measure for evaluating the surface quality of nylon after cutting.  Thera are a couple of questions:

1. The lambda criterion is clear and logical but since the whole study relies on it, there is a question: why did authors choose the "flat/bured" ratio? Is there any reference? Is it the only factor important to Nylon cutting?

2. Figures 8 and 14 have no a/b labels.

The manuscript delves into force analysis, offering theoretical insights that substantiate the advantages of ultrasonic cutting. 

The manuscript can be published if it conforms to the Coatings scope.

Author Response

First and foremost, we would like to express our sincere gratitude for the valuable comments and suggestions provided during the review of our manuscript titled
"Relationship between Nylon Filament Cut Profile and the Sharpness of Rotary Razors." Your feedback is instrumental in enhancing the quality of our research. Below, we address the reviewer's comments.

Comments 1: The lambda criterion is clear and logical but since the whole study relies on it, there is a question: why did authors choose the "flat/bured" ratio? Is there any reference? Is it the only factor important to Nylon cutting?

Response 1: The reviewer inquired why we chose the "flat/burred" ratio as the basis for our study and whether it is the only important factor in Nylon cutting. We selected this ratio based on observations of four phenomena (A, B, C, D) that occur when external force is applied to nylon cutting. We found that sections A and B exhibit lower surface roughness. To improve the end-face quality, a higher proportion of sections A and B is preferable, hence the introduction of the ratio λ=(C+D)/(A+B). The lower the λ ratio, the better the cutting quality. If λ were to be calculated as (C+D)/(A+B+C+D), it would not reflect the morphology of the end face of section A caused by extrusion deformation. Our choice is grounded in experimental observations and analysis, aiming to propose an intuitive and effective method for assessing the quality of nylon cutting. Future research may explore additional factors affecting the quality of nylon cutting.

Comments 2: Figures 8 and 14 have no a/b labels.

Response 2: We thank the reviewer for pointing out the omission in the figures. a/b labels have now been added to Figures 8 and 14 to clearly present the research results.

Additional clarifications
Once again, we are thankful for the thorough review and valuable comments from the reviewer. We look forward to any further guidance and suggestions. Should there be any questions or additional information required, please do not hesitate to contact us.

Reviewer 2 Report

Comments and Suggestions for Authors

The article is dedicated to the Evaluation and Characterization of Ultrasonic Cutting of Monofilament Nylon. It is written in good language and sounds scientifically. This manuscript is a study of the cleanliness of the cut of nylon monofilament under conventional and ultrasonic assisted cutting conditions and can be interested for specialists in the field.

Reviewer comments:

A reference to conventional nylon cutting should be added to the title as it has been researched to the same level of confidence.

The article does not explain why the frequency of 24.75 kHz was chosen. To complete the study, it would be nice to conduct a study of the influence of ultrasonic frequency on the cleanliness of the cut.

Figure 8, 14 (a)-(b) on the pictures itself are absent. On the Figure 14, the result of the conventional cut appears smoother, flatter and more uniform than the ultrasonic cut. Perhaps the pictures are mixed up.

Page 8 line 208 “microscopy was 25.2, 13.7, 26.8, 32.8 for segments A–D” units are missed.

Conclusion: Accept after minor revision.

Author Response

First and foremost, we would like to express our sincere gratitude for the valuable comments and suggestions provided during the review of our manuscript titled "Relationship between Nylon Filament Cut Profile and the Sharpness of Rotary Razors." Your feedback is instrumental in enhancing the quality of our research. Below, we address the reviewer's comments.

Comments 1: A reference to conventional nylon cutting should be added to the title as it has been researched to the same level of confidence.

Response 1: We appreciate your suggestion to highlight the comparison with conventional nylon cutting more prominently in our manuscript, including the introductory. Based on your recommendation, we have modified the introductory to better reflect the emphasis on both ultrasonic and conventional cutting methods. This adjustment aims to clarify the scope of our research and its comparative nature from the outset.

Comments 2: The article does not explain why the frequency of 24.75 kHz was chosen. To complete the study, it would be nice to conduct a study of the influence of ultrasonic frequency on the cleanliness of the cut.

Response 2: You raised an important point regarding the selection of the ultrasonic frequency used in our experiments. The designed frequency for our ultrasonic tool is 25kHz, with the actual working frequency being 24.75 kHz, which falls within the acceptable error range. We acknowledge the potential interest in exploring how different ultrasonic frequencies might affect the cutting cleanliness. However, our study's primary focus was on a comparative experiment between ultrasonic cutting and conventional cutting of nylon. As such, we did not investigate the influence of varying ultrasonic frequencies in this study. We believe this could be an interesting direction for future research and have noted it in the discussion section as a potential area for further exploration.

Comments 3: Figure 8, 14 (a)-(b) on the pictures itself are absent. On the Figure 14, the result of the conventional cut appears smoother, flatter and more uniform than the ultrasonic cut. Perhaps the pictures are mixed up.

Response 3: We have corrected the labeling in Figures 8 and 14 to include a/b labels for clarity. Additionally, we realized there was a mistake in the order of images in Figure 14, which may have led to confusion regarding the comparison of cut quality between conventional and ultrasonic methods. This has now been rectified, and the correct order of images should clearly illustrate the differences in cut quality, consistent with our findings.

Comments 4: Page 8 line 208 “microscopy was 25.2, 13.7, 26.8, 32.8 for segments A–D” units are missed.

Response 4: We apologize for the oversight in omitting the units for the microscopy results. The units have now been added to Page 9, line 226, ensuring the results are presented accurately and clearly. 5.

Additional clarifications
Once again, we are thankful for the thorough review and valuable comments from the reviewer. We look forward to any further guidance and suggestions. Should there be any questions or additional information required, please do not hesitate to contact us.

Reviewer 3 Report

Comments and Suggestions for Authors

Manuscript Number: coatings-2959179

Title: Evaluation and Characterization of Ultrasonic Cutting of Monofilament Nylon

Decision: Minor revision

Article Type: Original article

The article is, in general, well written but there are some issues that article should consider to revise in order to improve its quality. Some comments were done in this way:

Ø  "We" should be avoided when writing articles. Please fix it. Do not use articles older than 10 years. To prevent these problems, we conducted experimental research comparing ultrasonic and conventional cutting of monofilament nylon and used the controlled variable method to analyze the micromorphology after cutting at different speeds and tensions.

Ø  Include more articles in the Introduction section. Also indicate the importance of this study in the last paragraph. Emphasize the difference from similar studies in this section.

Ø  Fig.7 should be edited.

Ø  In addition to the topography images given in Fig 14, add surface roughness values such as Ra, Rz, Rt, Rp for all parameters.

Ø  Make the scale larger in Fig 14.

After the corrections have been made, I ask that they be sent to me for review.

Author Response

First and foremost, we would like to express our sincere gratitude for the valuable comments and suggestions provided during the review of our manuscript titled "Relationship between Nylon Filament Cut Profile and the Sharpness of Rotary Razors." Your feedback is instrumental in enhancing the quality of our research. Below, we address the reviewer's comments

Comments 1: "We" should be avoided when writing articles. Please fix it. Do not use articles older than 10 years. To prevent these problems, we conducted experimental research comparing ultrasonic and conventional cutting of monofilament nylon and used the controlled variable method to analyze the micromorphology after cutting at different speeds and tensions.

Response 1: We have carefully revised the manuscript to avoid the use of personal pronouns, as suggested. The revisions include changes on Page 1 (lines 8-9), Page 2 (lines 59-60, 76), Page 3 (line 106), Page 5 (line 150), Page 6 (line 172), Page 7 (lines 197, 200), and Page 10 (line 236). Additionally, references older than 10 years have been replaced with more recent studies to ensure the relevance and currency of the literature review: Page 1, line 28-44.

Comments 2: Include more articles in the Introduction section. Also indicate the importance of this study in the last paragraph. Emphasize the difference from similar studies in this section.
Response 2: In response to your valuable suggestion, we have enriched the introduction with additional recent articles that further contextualize our study within the current research landscape. (Page 1, line 28-44.) The final paragraph of the introduction now explicitly highlights the significance of our study and delineates its unique contributions compared to similar works. (Page 2, line66-74.)

Comments 3: Fig.7 should be edited.

Response 3: Figure 7 has been edited to improve its clarity and informational value in line with your recommendations.

Comments 4: In addition to the topography images given in Fig 14, add surface roughness values such as Ra, Rz, Rt, Rp for all parameters.

Response 4: We have augmented Figure 14 to include surface roughness values (Ra) alongside the topographical images for a more comprehensive representation of the cutting effects under different parameters.

Comments 5: Make the scale larger in Fig 14.

Response 5: The scale in Figure 14 has been enlarged for better visibility and to facilitate a clearer interpretation of the findings.

Additional clarifications

Once again, we are thankful for the thorough review and valuable comments from the reviewer. We look forward to any further guidance and suggestions. Should there be any questions or additional information required, please do not hesitate to contact us.

Round 2

Reviewer 3 Report

Comments and Suggestions for Authors

The authors have made the desired changes. The article can be published in this form.